# Assembling Frameworks for Strategic Innovation Enactment: Enhancing Transformational Agility through Situational Scanning

**Joel Bigley**

Jabs School of Business, California Baptist University, Riverside, CA 92504, USA; jbigley@calbaptist.edu

**Abstract:** Although a significant body of knowledge has been created around strategic management that drives change and innovation, there are voids in the literature regarding assembling a flexible and localized environmental scanning (ES) framework needed to assess threats, opportunities, the current environment, and the desired end-state of a global value chain (GVC) with direct linkages to strategic enactment. In this paper, the author describes this gap and develops a capability to create a task plan from a situational ES. The need for ES customization is further validated as a result of the change during a transformation or adaptation, hence the need for a localized feedback mechanism.

**Keywords:** environmental scan; situational agility; global value chain; localized strategy; strategic agility

## 1. Introduction

This article attempts to provide enhancements to theoretical models while exploring the establishment of situation driven frameworks for environmental scanning (ES) and future scanning (FS). Taking a phenomenological view of the existing theory, the author posits that innovation is local, as markets are local and motivational interest is local (Khan et al. 2016). Studies have shown that higher-decision-making autonomy increases the probability of a subsidiary, in an internationally distributed innovation network, developing product innovation (Beugelsdijk and Jindra 2018). Innovation may also be shared across linkages in an internationally distributed 'innovation network'. These linkages may take on the form of strategic alliances, coopetition, network contracts, and other innovative constructs (Venturelli et al. 2018). According to (Li et al. 2018), a firm's innovativeness is related to its connection with political and economic external stakeholders. Other research emphasizes the liberation of innovation by ending relationships with suppliers who are a source of inertia (Zaefarian et al. 2017). Inertia is a counterforce to strategic agility. Fluid change is mandated by a demanding business environment. "Business agility is the ability to swiftly and easily change businesses and business processes beyond the normal level of flexibility to effectively manage unpredictable external and internal changes" (Van Oosterhout et al. 2006, p. 132). Several analogies are apropos. In athletics, agility is the ability to change speed or direction quickly while anticipating the moves of competitors around you (Gabbett et al. 2008). In IT infrastructure, the ability to change is critical to configuration changes that are inevitable (Hoogervorst 2004). Regardless of the scenario, strategic agility is critical to surviving and thriving in any market.

With this in mind, there is a gap between theory and success when it comes to enacting strategic plans based on the discovery process that describes environmental scanning. This theoretical study compares the existing theoretical posture of two theories, ES and strategic enactment, and connects them, thereby enhancing the existing theory. In fact, research shows that strategic plans for achieving innovation in global value chains (GVCs) usually does not live up to expectations, because

environmental scanning, and, therefore, strategic planning, is in a state of disarray characterized by conflicting definitions, blurred boundaries, confused measurement methods, and conflicted findings (Brackertz and Kenley 2002; D'Aveni et al. 2010; Jogaratnam 2005; Kono and Barnes 2010; Love 2011). Studies have also shown that an ability to sustain a competitive advantage continues to diminish over time (Ruefli and Wiggins 2003). This is further exacerbated by the increased volatility of financial returns (Thomas and D'Aveni 2009). Environmental scanning is critical to innovation strategic enactment, and so, practitioners need to better understand how accurate situational assessments are enabled by customized scanning frameworks. The objective of the article is to fill the gap between the existing theory and a new perspective that bridges the gap between ES creation and strategic enactment. This position is supported by the knowledge that socio-cultural elements are critical business success factors; even so, they are often ignored (Harding and Rouse 2007; Shimizu et al. 2004).

Since its conceptualization, a body of knowledge has accumulated around strategic innovation (Aguilar 1967). There is, however, the need for a better understanding regarding assembling a situation-based scanning framework needed to assess both a current reality and a desired end-state. Concepts related to (1) the formation of customized scanning frameworks using existing and created domains, (2) planning within frameworks exploiting relationships between dependent domains, and (3) the deployment of domain-specific planning focused on a performance gap will be discussed.

Organizational theorists are interested to classify and describe environmental attributes (Bourgeois 1980); however, capability to create a customized framework is generally avoided in favor of specific framework designs with varied applicability (CIPD 2013; Cornelissen 2004; Leeman 2010; Schmieder-Ramirez and Mallette 2007). Given that an environmental system may be too complex and interdependent to be analyzed as a whole, it must be disaggregated and assessed in terms of its individual components for better understanding (Boyd and Fulk 1996; May et al. 2000; Sawyerr 1993). In addiditon, consideration should be given to the possibility that domains, which make up ES frameworks, may change or drift during a transformation (Daft and Weick 1984; Hambrick 1982). Furthermore, where shared variables are present, cross-domain, cross-functional, and cross-hierarchical scanning is a factor (Hambrick 1981). Complexity is added from linkages between scanning methods, environments chosen, and strategic orientation (Jennings and Lumpkin 1992). It is clear, therefore, that an accurate awareness of a situation through an evaluative framework is critical to task selection and prioritization (Hambrick 1982), helping leaders navigate obstacles (Bossidy et al. 2002; De Pree 2004). Conversely, even though some objective environmental attributes may differ (Bourgeois 1985; Snow 1976; Starbuck 1976), narrow, parochial, or superficial views are not adequate to handle the complexity of scanning (Dess and Beard 1984; Slaughter 1999).

## 2. The Argument

It will be illustrated that (1) there is a holistic and systemic method (Linstone and Mitroff 1994) for domain, or environment, selection that best matches an organization's situation, (2) that there is a library of domains for use that has already surfaced in the industry (Costa et al. 1997; Schmieder-Ramirez and Mallette 2007), (3) that domains can be created from data analysis, (4) that there is connectedness within an environmental framework through dependencies between relevant domains, (5) that capability to select relevant domains is critical to strategic efficacy, and (6) that there is a relationship between domain selection, sometimes referred to as value domains (Doz and Kosonen 2008), and the desired business outcomes. It has been established that planning tools and methods are needed for 'dynamic maneuvering' (El Sawy 1985; Grimm et al. 2005). This article discusses how a flexible framework fits into a strategic planning roadmap, aiding practitioners in strategic enactment.

A number of key questions are discussed in this article, including the following:

- How would a situational framework be constructed and exploited?
- How are environmental domains interdependent?
- How can a framework create new advantages and avoid self-cannibalization?

- And finally, how are endogenous and exogenous aspects of a domain accommodated?

*Explication of Research Questions*

To explain the rationale of the questions, the following should be considered. While environmental scans are executed, they are often not exploited. As a result, the exercise does not create value. Part of the issue with the scan is that the domains are not appropriately defined in scope or context. This article will discuss how to create the domain within an appropriate context. Aside from the appropriateness of the domain, what often fails to be recognized is the interdependency between domains. In some cases, the relationship is stronger than others. A network mindset should be used when considering the domain set. The purpose of the scan should be action and influence through understanding. With this awareness, strategic plans should be executed that deal with external interventions. Some of these cannot be anticipated, however, a robust risk management posture may be able to thwart the risk loss associated with known and unknown risk potential. Again, based on dependencies, some thought needs to be applied to relationships between domains. One domain may benefit from an action while another is hurt. In some cases, a self-cannibalization effect occurs. Leaders who are aware need to decide if it is worth it. Finally, an environmental scan frequently is only concerned with external threats and opportunities, while ignoring internal opportunities. These internal opportunities offer up advantages to a firm that knows and exploits them.

To deal with these questions, the article is organized as follows: The first section discusses aspects of environmental scanning. The second section discusses scanning variables, a vital component usually minimized for expediency. A third section discusses the applications of frameworks in customizable scanning activities, including variable driven metrics within dynamic domains and sub-domains. The fourth section discusses forces involved within domain-based initiatives and their associated metrics. The fifth section discusses strategic enactment and how frameworks fit into the strategic plan. Finally, there are the implications and a conclusion, including a list of concepts recommended for further research.

## 3. Part I. Environmental Scanning

Environmental scanning (ES) is a system allowing information about an organization's external environment to be collected and exploited for strategic purposes (Albright 2004; Choo 1999; McEwen 2008; Yasai-Ardekani and Nystrom 1996). It applies to the existing environment and to an anticipated environment. Future scanning (FS) uses early warnings that help managers develop a planning horizon (Aguilar 1967; Choo and Auster 1993). FS has been primarily directed outwards ('out there'), while, typically, inward looking trends ('in here') are often ignored. Furthermore, dependencies between environments are often not understood (Slaughter 1999), resulting in a lack of real progress along strategic roadmaps (Brackertz and Kenley 2002). This perceived progress driven by the illusion of action may result in a zero-net gain.

### 3.1. Types of Scanning

Competitive scanning is linked to market opportunity and threats from the competition (entities, technology, megatrends, etc.), or other influences (Doz and Kosonen 2008). Competitive advantage comes from obtaining valuable industry foresight (Hamel and Prahalad 1994); however, environmental hostility, in the form of intense competition, overwhelming climates, market commonality, resource similarity, and a lack of exploitable opportunities, threatens an organization's ability to achieve their desired business outcomes (Covin and Slevin 1989; D'Aveni et al. 2010). An organization's competitive attitude and capability within its environments should be understood (Alvesson and Lindkvist 1993; Brown and Starkey 1994; Deming 1986; Peters and Waterman 1982; Rousseau 1990; Senge 1990).

The frequency of scanning activities tends to increase with environmental uncertainty; however, it decreases when uncertainty is overwhelming, when absorptive capacity is exceeded (Choudhury and Sampler 1997; Cohen and Levinthal 1990), or when useful information is not accessible

(Hough and White 2004; May et al. 2000). Why? Because it does not produce meaningful results. Concurrently, a perception of diminishing returns from scanning efforts in a stable environment may lull an organization into catatonic complacency (Hough and White 2004) or entropy (D'Aveni et al. 2010; De Pree 2004). In this way, the organization foregoes an aggressive competitive stance. This posture may also concede a competitive position.

Informal, or ad hoc, scanning by middle- and top-level executives is typically short term, infrequent, fragmented, and may be initiated by a crisis (Aguilar 1967; Hambrick 1979; Hambrick 1981; Kefalas and Schoderbek 1973). Even though managers below top executive levels typically conduct scanning more frequently (Hambrick 1981), strategy making is linked to subjective interpretations in difficult to comprehend and rapidly changing environments (Elenkov 1997; Jogaratnam and Wong 2009; Hambrick 1981). A proactive stance is further inhibited when top executives assume that lower level executives are performing scanning, when in fact they are not (Hambrick 1981).

*3.2. Scanning Success Factors*

Scanning accuracy is dependent on the domains selected and the approach taken (Hrebiniak and Joyce 1985). For example, the accounting/finance domain might include clear-cut scanning behaviors, while a process engineering function without clear role definitions or explicit bounds might have a more tenuous linkage to scanning techniques (Hambrick 1981). Perception accuracy is a basis for managerial action (Tsai et al. 1991), and scanning is the first step in the development of perceptions (Carpenter and Fredrickson 2001; Davis and Meyer 1998). Consequently, scanning voids are particularly risky (Hambrick 1981), jeopardizing needed adaptation. It is clear that continuous scanning must include structured data collection using optimized frameworks that clarify perceptions, tasks, and reveal actual results (Bourgeois 1985).

Two general measures of scanning strategy are frequency and scope (Beal 2000; Yasai-Ardekani and Nystrom 1996). The scope of a scanning effort may include industry competitor analysis, marketing research, market-organization alignment, consumer analysis, new product development, supplier capability, and service innovations (Daft and Weick 1984; Grant 1998). Scanning frequency may be driven by a transitional economy, regulatory shifting, and demographic or cultural trends (Asheghian and Ebrahimi 1990; May et al. 2000; Schneider and Meyer 1991). The range of characteristics of an expected environment helps leaders make decisions today that align them with a desired future, at a suitable pace.

In high performing organizations, scanning frequency, scanning intensity, and scanning type (Jennings and Lumpkin 1992) matches, or exceeds, the market change rate (Choudhury and Sampler 1997), so that desired future states can be realized in time (Hough and White 2004; Sawyerr et al. 2000). A lack of predictability, market fluidity, and environmental complexity drive scanning efforts (Czarniawska 2007; Duncan 1972; Dutton and Jackson 1987; Ebrahimi 2000; Thompson 1967). Effective information synthesis is driven by environmental dynamism, the rate of innovation, munificence, (Castrogiovanni 1991; Duncan 1972), technology, globalization, industry convergence, aggressive competitor behavior, an increased frequency of temporary advantages, fleeting equilibrium, and general uncertainty. This then promotes the need for the organization to synchronize with relevant environments (Aldrich 1979; Bluedorn 1993; Freel 2006; D'Aveni et al. 2010; Richard 2003).

Leaders in hostile and non-munificent environments are especially challenged to comprehend external threats (Anderson and Tushman 2001; Fahey and Narayanan 1986; Goll and Rasheed 1997; Snyder 1981). Firms perceive their environments differently. This depends, at least partially, on their strategic approach, and if data is involved (Zahra 1987). Business intelligence influences strategic decision making. Data completeness and analyzability influences sense-making (Sutcliffe 1994). Proactively, data structures must assist with the processing needed to develop, pursue, and monitor a strategy (Choo 2001; Jogaratnam and Wong 2009; Lau et al. 2012). Otherwise, leaders may decide an environment is unanalyzable, avoiding it at their own peril (Aguilar 1967; Ferrier et al. 1999).

Ultimately, sense-making occurs when leaders construct an environment by framing experiences and by creating new capabilities (Milliken 1987; Weick 1987). Strategic enactment occurs when activity, often simultaneous, is introduced to accomplish tasks, create new capabilities, and create sense within them. It is clear then that managers with a limited capacity for information processing have to be efficient in their approach (Daft et al. 1988), in order to get a predictive picture of what is to come; hence, a need for scanning accuracy.

*3.3. Scanning Variables*

Leaders comprehend scanning results when they understand interaction between the domain variables and their influence on performance (Aguilar 1967; Hambrick 1979; Kefalas and Schoderbek 1973; Venkatraman 1989). Attributes of variables in environmental scanning could include a hierarchical level, specialty level, personality dimensions, environmental complexity, rate of change, firm size, impact and frequency of risk events, and information source reliability (Jennings and Lumpkin 1992; Lindsay and Rue 1980; Robinson 1982; Valencia 2010). The need for these variables, their variety, a variation range within each one, and their weighting validate the need to customize a scanning framework to a specific situation.

The literature categorizes variables as controllable (e.g., location, pricing, customer base, market penetration, management, throughput, and capacity allocation) and uncontrollable (e.g., consumer behavior, competition, technology, innovation, economic conditions, seasonality, legal or regulatory restrictions, and market evolution) (Mitroff and Emshoff 1979). Controllable variables can be influenced, while uncontrollable variables typically force adaptation (Mitroff and Emshoff 1979). Porter (2008) presents a model to obtain external competitive data; however, variables may be market centric. For example, the hospitality industry may be more tuned to variables such as competitor market share, market saturation, sales stagnation, margin erosion rate, rate of consolidation, and failure rates (Tse and Olsen 1999). Additionally, directly and indirectly, variables have endogenous and exogenous characteristics (D'Aveni et al. 2010; Thoumrungroje and Tansuhaj 2005). A measuring system needs to accommodate these attributes and accurately represent the variables chosen. Significance mapping tools can help a practitioner understand value, associations, and trait influences of critical performance indicators. Of course, data collection planning and analysis methods assure that the data collected is complete, relevant, and timely (Choudhury and Sampler 1997). Finally, statistical analysis, using key domain variables, can be used to provide inferential conclusions from within a scope of a selected or created domain.

Variables may be dependent on the strategy selected by managers. For example, a cost cutting strategy would promote variables related to cost, competitor innovation, and efficiency. A sales focus strategy would include market penetration tasks in an industry segment. A differentiation strategy would suggest variables centered on product superiority and attributes (Hill 1988; Hrebiniak and Joyce 1985; Jennings and Lumpkin 1992; Miller 1988, 1989; White 1986). Furthermore, a strategy may be proactive, looking for measureable opportunities, or reactive, looking for multidimensional problems or threats (Ansoff 1975; Mintzberg 1973). Additionally, variable selection could also be influenced by the impact of risk-taking propensities, culture continuity, risk appetite and tolerance, tolerances for ambiguity, and the influence locus of control (Anderson and Frigo 2012; De Pree 2004; Miller and Toulouse 1986); regardless, continuous strategy innovation, including variable selection, is necessary in turbulent high-velocity markets (D'Aveni et al. 2010). In sum, variables describe domain characteristics prior to and following transformation related initiatives. An appropriate collection of variables informs planning task lists that need to be executed.

## 4. Applying Customizable Frameworks

### 4.1. Domain Library

Strategic planning starts with an accurate description of a current situation (Davis and Meyer 1998). Domains should be chosen that are a best representation of the current reality. Once a baseline condition is thoroughly understood, a compelling vision can be formed. While an accurate framework can be built using existing and created domains, unfortunately, scanning using equally weighted domains may result in an intolerable error percentage. Some frequently used domains from literature include Engineering, Administration, Security, Social, Accounting/Finance, Political, Operations, Economic, External Growth, Product Development, Legal, Security, Regulatory, Marketing, Cultural, and Technology. (Hambrick 1981; Kefalas and Schoderbek 1973; Schmieder-Ramirez and Mallette 2007). If deployed accurately, a domain assembly, or framework, can produce a more accurate 'as is' assessment and a more usable gap analysis against a desired 'to be' state. A more efficient roadmap can then document the transition.

### 4.2. Domain Creation

Qualitative data from scanning is typically language based (Crowston et al. 2012). Additionally, business intelligence includes both internal and external data (Lau et al. 2012) that must be exploited. It follows then that, after the data is categorized and coded, theme clusters emerge. These theme clusters, which are prominent forces in the scenario, may be eligible for domain status if their boundaries can be described. They, together with other domains selected from the library, can be assembled into a scanning framework. Relationships between themes can be understood using available tools such as the Affinity Diagram and the Interrelationship Diagram. This promotes knowledge around dependencies, including positive or negative relationships. These tools can also be used to assign significance to each domain and to its connections. Understanding the ways that themes connect informs knowledge about strength and the behavior of network connections.

### 4.3. Domain Selection

Domain selection can be accomplished using a variety of common methods, and may include active listening. Brainstorming activities, using a full range of stakeholders from executives to those working 'at the coal face', reveal a wide range of relevant domain labels along with their applicable characteristics. The significance of the domains and their characteristics can be captured using a tool such as nominal group technique (NGT). A list of domains offered up can be associated with the industry or be general across all of the industries. Where there is a need, domains can be created; however, new domains require an appropriate label, a description, characteristics, variables, and a domain to sub-domain breakdown. A domain description should also include a scope statement so that 'the edges' are clearly defined.

### 4.4. Metrics Tree

Situation based scanning frameworks should be deployed with desired business outcomes in mind (Kono and Barnes 2010). Each sub-domain has business outcomes that, when aggregated, reflect the outcomes for the domain in general, as illustrated in Figure 1 below. A balanced scorecard (Kaplan and Norton 1996) reflects the measurements from variables selected to represent all of the domains present. Sub-domains are described numerically through the use of variable based metrics. These metrics are broken down into sub-metrics for clarity and specificity. Granularity provides managers with knowledge about where they are and accurate feedback about the impact of their actions.

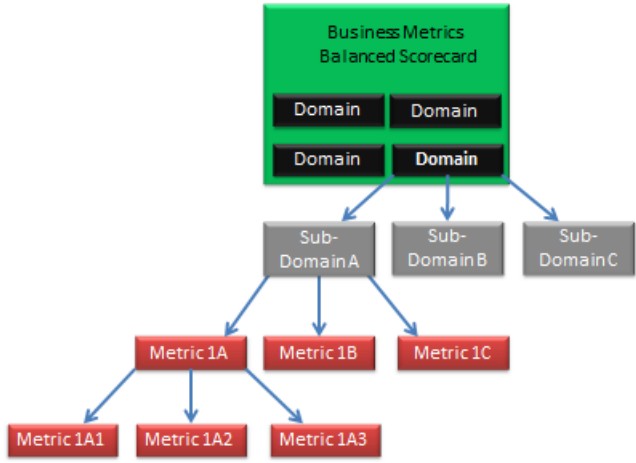

**Figure 1.** Domain metric break down.

## 4.5. Domain Structure

The ability to divide domains into sub-domains helps break down the influential aspects of an environmental segment. For example, a legal domain could be broken up into any combination of city law, state law, federal law, and organizational policies. These are inter-dependent as they cannot be inconsistent or conflicting. Within each sub-domain, key performance indicators, strategic tasks, force field analyses, and task weighting based on impact and urgency, can be documented. Force field analyses can be used to position strengths to seize opportunities and to identify weaknesses and threats for risk mitigation. Tasks derived from these analyses become building blocks for an effective plan that produces desired outcomes. This is illustrated further in Figure 2.

| Environment | Sub-Environment | Goal | Driver | Positive Effect | Negative Effect | Action ID | RM | Action Description |
|---|---|---|---|---|---|---|---|---|
| Domain | Sub-domain A | Goal A | Driver A | Positive 1A | Negative 1A | AP 1A1 | 50 | |
| | | | | | | AP 1A2 | 20 | |
| | | | | | | AP 1A3 | 10 | |
| | | | | | | Residual1A | 20 | |
| | | | | Positive 2A | Negative 2A | AP 2A1 | 30 | |
| | | | | | | AP 2A2 | 10 | |
| | | | | | | AP 2A3 | 10 | |
| | | | | | | Residual2A | 50 | |
| | | | | Positive 3A | Negative 3A | AP 3A1 | 80 | |
| | | | | | | AP 3A2 | 10 | |
| | | | | | | AP 3A3 | 5 | |
| | | | | | | AP 3A4 | 5 | |
| | | | | | | Residual3A | 0 | |
| | | | Driver B | Positive 1B | Negative 1B | AP 1B1 | 30 | |
| | | | | | | AP 1B2 | 30 | |
| | | | | | | Residual1B | 40 | |
| | | Goal B | | | | | | |
| | Sub-domain B | | | | | | | |
| | Sub-domain C | | | | | | | |

(Gantt)

**Figure 2.** Force field analysis to Gantt.

A domain description limits the selection of metrics to those that are applicable within a domain scope. An understanding of the variables involved can help with metric descriptions, including the mathematical logic applied to the calculation. A resultant suite of metrics, when applied to a current situation, can describe the 'as is' state. These metrics can also be used for benchmarking purposes. A SWOT analysis within a domain will reveal strengths, weaknesses, opportunities, and threats (SWOT), as illustrated in the taxonomy in Figure 3. Once these attributes are thoroughly understood, management can determine the performance gap. Objectives around gap closure should be described quantitatively. Disparity between the current and desired state leads to a roadmap that, when enacted and fulfilled, will accomplish the organization's goals. It follows that a roadmap must be aligned with

the vision (Kaplan and Norton 2006). Residual objectives, those that have not been completed, should be listed and addressed at a later time. If they are not retained, they may be forgotten along with any potential opportunities associated with them. In sum, by breaking down framework elements and capturing the tasks needed to close gaps, a thorough action plan can be assembled, tracked, and monitored.

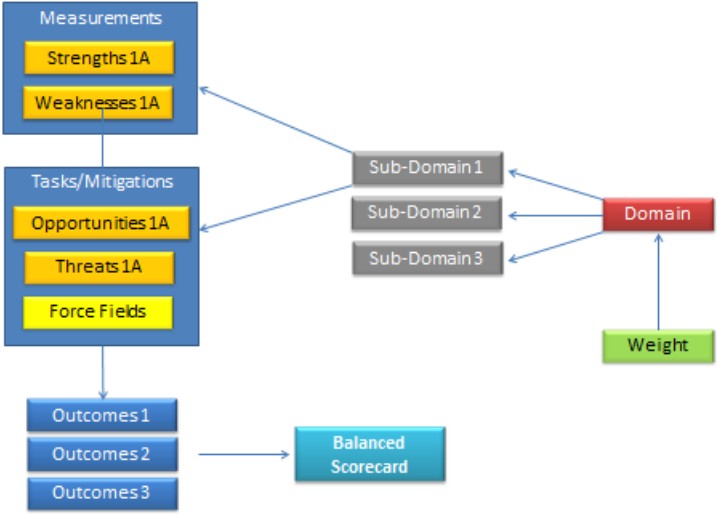

**Figure 3.** Domain taxonomy.

### 4.6. Domain Significance

A domain is weighted relative to the influence of the other domains in the framework. A dominant outcome driver, or dominant domain, should not be ignored or treated as an equal. Domain weights can be assigned using a Likert scale, or be linked to variable significance. Domain specific tasks and their weights inform the overall strategic plan. An understanding of the dynamic nature of internal and external metrics (Bandy 2002), a prospect of future expectations (Chrusciel 2011), and an awareness of the weighted performance drivers on the critical path are essential to the strategic plan.

### 4.7. Internal–External Dynamics

Domains may be chosen for their inward and outward attributes that influence the network. For example, a 'legal' domain has endogenous and exogenous aspects. Inwardly, policies and rules apply to local environment controls and drives behaviors. Outwardly, laws are imposed on clients regarding fulfilling contractual obligations, as an example. Either of these influences can affect other domains in the network. For example, when legal clauses enforce contractual payment terms, this will impact the financial domain in a positive way.

Domain and sub-domain characteristics relate to domain function and explain the dynamic that exists within its scope. A 'legal' domain, again for example, can be broken into sub-domains, as illustrated in Figure 4. In this example, a domain is broken into 'internal' and 'external' attributes based on actions. Internally, a legal domain accomplishes actions for others and imposes actions on others. Externally, a legal domain is needed by others and it is the interface to others.

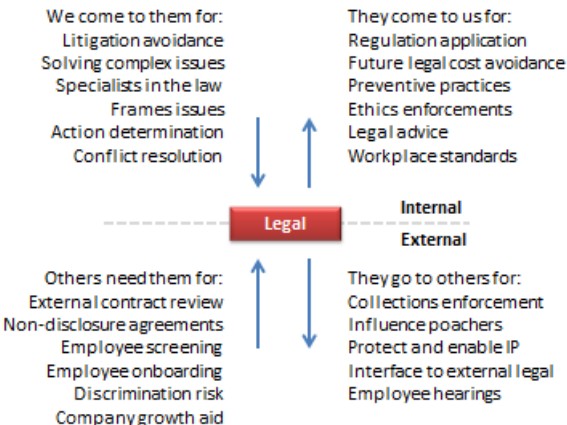

**Figure 4.** Domain internal and external attributes.

*4.8. Connectedness between Domains*

A framework should be viewed as a network. An action in one domain can trigger a positive or negative reaction within a network. This is, at least partially, analogous to a spider sensing an opportunity or threat in its web. Typically, a 'silver bullet' approach to problem solving ignores either a ripple effect or the limited impact within the connected nodes, as illustrated in Figure 5. This can be beneficial to the network. Conversely, if an appropriate selection of influential domains is assembled into a framework, the system's sensitivity is optimized. Irrelevant domains have been excluded, reducing the noise in the system that impedes usable feedback. Unnecessary variables add unnecessary complexity. Ideally, increased system responsiveness and predictability are desired following or during a change. The complexity and dynamism of this network should not be underestimated, and so, a system approach is necessary (Dess and Beard 1984).

Environmental dynamism is uncertainty from instability, variability, and volatility within a current reality (Davis and Meyer 1998). Change is difficult then, due to a lack of predictability (Dess and Beard 1984). In some cases, there is a higher dependency between the domains than expected. While one domain may have positive connection strength to another, there may be a situation where a domain has an inverse influence on another domain. For example, an application of a technical solution to increase a security domain may negatively impact a social domain, as people resist and rebel against the imposed change. A deployment of a solution may have been a task on the strategic plan; however, actions are absent that deal with corresponding resistance, leading to task failure. This suggests that a domain may be isolated within a framework; however, not understanding dependencies between the domains compromises an awareness that is complete and threatens the execution of planned tasks.

To illustrate the inter-connectedness further, using a familiar object, a soccer ball is a truncated icosahedron that must survive the impact of being struck by an edge on a soccer boot. A reaction to this impact must be predictable so that a soccer player can direct a ball towards a destination. Air pressure from inside the ball provides an opposing force, as connected patches on the ball's surface have to withstand the forces placed on them internally and externally. These forces are not balanced, or evenly spaced. So, it is, to some extent, for organizations that suddenly have to go through change, expected or not, and then assume a temporary equilibrium before the next change event. If the structure has strong links to bear the force of the change influence at any point in the organization, it will be resilient to change (Leflar and Siegel 2013). If one of the connectors is weak, the structure is at risk of collapse (Wilber 1995).

*4.9. Internal vs. External Influences*

Organizations need to respond proactively to external changes and avoid surprises (Davis and Meyer 1998). This response capability relates to avoiding surprises internally as well as externally. While the voice of external customers is typically heeded, the voice of internal customers is often ignored. With resources that are imperfectly fluid, non-tradable, non-substitutable, inimitable, rare, or of decreasing value, internal customers' expectations are met less frequently, resulting in frustration. Additional endogenous antecedents could include the impact of internal decisions, competitive actions, and cultural behaviors that undermine an organization's competitive advantage (D'Aveni et al. 2010). As an example, constructive feedback to improve performance needs to be readily available, interpreted, and then exploited (Sharfman and Dean 1991). If ignored, a turnover rate trend may make an organization incapable of achieving higher revenues, because of its negative influence on capacity, talent availability, or agility. When this occurs, an organization is more sensitive to small changes in the external environment and it is increasingly difficult for an organization to focus on performance or change (Chrusciel 2011).

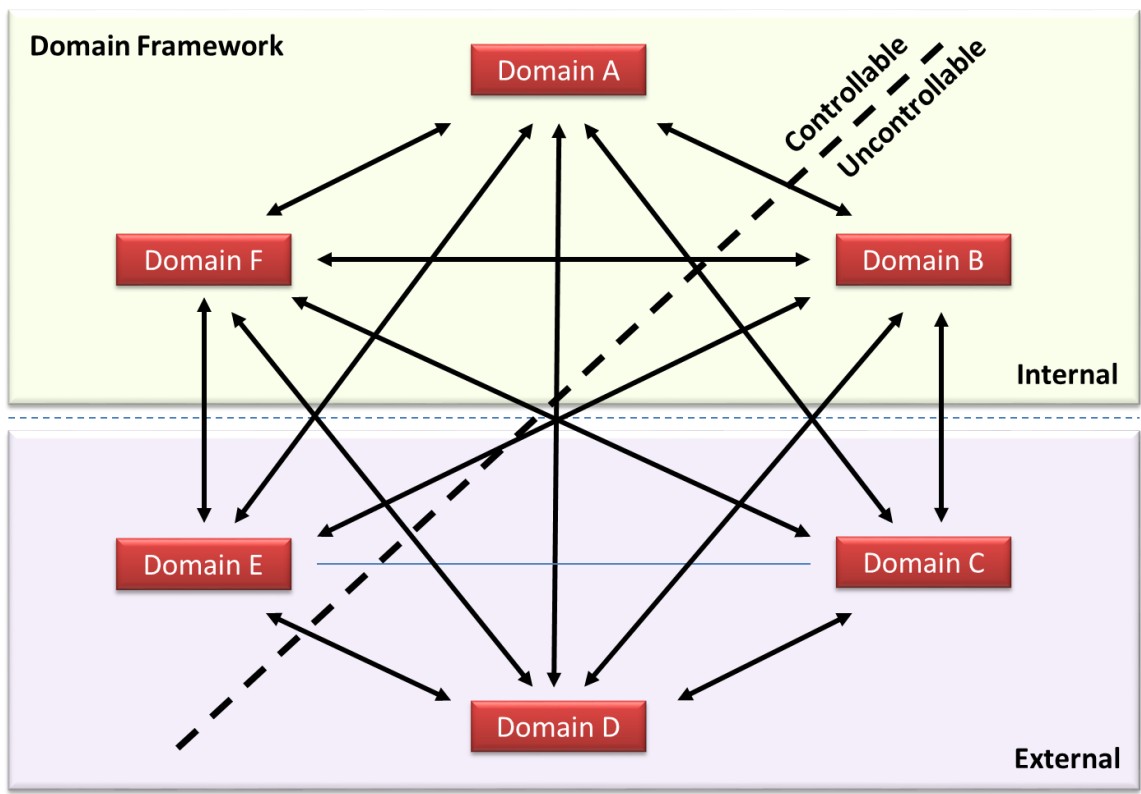

**Figure 5.** The domain framework.

## 5. Part II. Force Field Analysis

Within a sample domain and sub-domain, goals are listed that relate to the vision, or the 'to be' state. A goal inherently includes drivers that support and drivers that detract from the potential achievement of the goal. Capturing positive and negative drivers within the domains creates understanding. Rationalizing the existence of negative influences for each positive driver, and the associated positive influences on each negative driver, are critical for task creation. For example, a positive driver might be 'a more hardened infrastructure.' The negatives experienced by participants within the scope of the action may include an increased usage of time for system authentication. Consideration should be given to minimizing the negative drivers so that the benefits of positive drivers dominate the scenario. Positive solutions need to be detailed, holistic, and robust, in order to

bring value. A pull or a created change of gravity helps influence the transformation momentum, as organizations move towards a recognizable and legitimate goal (De Pree 2004).

*Negative Influence and Mitigation*

Essential to the analysis, a negative driver can have diminished influence by applying complimentary positive drivers as a counter force. A task list, as shown in Figure 6 below, must include actions that optimize opportunities for success. Each task associated with countering negative drivers should be thoroughly described along with the associated impact. A risk mitigation (RM) column provides a place for this. For example, adding quick biometric authentication for system users might contribute to making the frequency of the authentication requirement less tedious for an operator to comply with. It may also reduce the stress associated with password management. Moreover, as operators are closest to the task environment, they are critical for obtaining opportunity and threat information, because this information is easily understood by them (Fahey and Narayanan 1986).

A force field analysis aids in task list creation, which leads to the achievement of desired business outcomes. While forced change encourages resistance, managing negative drivers helps with plan acceptance, because of a perception of mutual benefit (Fisher et al. 1991). It should be noted that a negative driver may require multiple positive actions to counter its opposing force. Again, a 'silver bullet' approach, where only one action is associated with achieving a goal, falls short and encourages change resistance because of imminent failure. As a force-field analysis usually does not mitigate all risk, a case may be made that some actions are undercompensated for other negative drivers, and either a sacrifice will need to be made or an arbitrary agreement will need to be lived with. Finally, tasks can be organized into a project planner, such as a Gantt, for scheduling, tracking, and resource planning. These tasks are executed, leveraging the available organizational slack or an external resource.

## 6. Part III. Measurements

To monitor the progress, unique characteristics of each domain and the achievement of domain specific business outcomes are measured. Sentiment analysis facilitates an extraction of wisdom from opinionated expressions during scanning (Lau et al. 2012). Unfortunately, defining the environment is difficult when executives describe them in terms of events or trends, rather than the characteristics or the sentiments of the participants (Hambrick and Snow 1977). Adding to the misrepresentation, managers define phenomena considered relevant, important, and desirable, based on their beliefs and expectations (Jennings and Lumpkin 1992). Furthermore, only some of the phenomena present in environments under scrutiny are included, because of constraints on a scanner's field of vision (Hambrick and Mason 1984), or because of an information overload and processing limitation (Farhoomand and Drury 2002; Lau et al. 2008; Yan et al. 2011). Managers also tend to formulate strategies, based on perceptions that deal with their unique situations (Goleman 1985; Starbuck 1976). This challenges a measurement system attempting to link key performance indicators and benchmarking data to manager perceptions (Frei and Harker 1999; Gilleard and Yat-lung 2004; Maiga and Jacobs 2004; Peschiera 2004; Tsang et al. 1999; Wauters 2005; Weick 1979). To partially remedy this, strategic planning should occur within each domain selected and must be relevant to the situation at hand, including domain specific business intelligence and measurements (Choudhury and Sampler 1997).

## 7. Part IV. Social Dynamics

The influence of organizational social dynamics, including enablers and inhibitors like momentum and inertia, are frequently underestimated and overlooked (Dobrev et al. 2003; Slaughter 1999). Momentum is an energy level driving change activities, coupled with the rate at which change is happening, while inertia relates to forces that reduce momentum and move an organization toward stagnation, mediocrity, rigidity, and entropy (D'Aveni et al. 2010; De Pree 2004; Doz and Kosonen 2008). These forces also relate to action progress within domains, and describe connections between the

domains, where dependencies exist (Boyd and Fulk 1996). While organizational flexibility is desired in order to sustain advantage, suboptimal framework constructs introduce obfuscation and ambiguity, compromising the ability to focus on roadmap execution. Framework design failures also introduce the unexpected; changing the task selection, sequence, and priority, leading to potential frustration, initiative fatigue, complacency, and despair among stakeholders (Audia et al. 2000).

## 8. Part V. Strategic Enactment

Selecting leadership for change activities is critical (Bossidy et al. 2002; Heifetz and Heifetz 1994; Smith et al. 2001; Wilkinson 2006). This leader needs to be an articulate and enthusiastic conceptualizer who is good at grasping strategies and explaining them (Bossidy et al. 2002). Leadership includes the prioritization, deployment, and measurement against established goals. If outcome measurements indicate that effort has fallen short of a target, a leader may initiate a limited improvement cycle as a remediation. Additionally, a framework review may be prudent because of the project duration and market turbulence. A framework conceived during a time of stability may not be applicable during or following a time of volatility (D'Aveni et al. 2010). Once the framework design has been fine-tuned and verified as being appropriate, a change leader should initiate a repeat scan to refresh the gap analysis data. A full strategic cycle is illustrated in Figure 6 below. Continuous improvement is an aggressive leadership activity, allowing an organization that embraces learning to keep pace with a rapidly evolving environment (Ferrier 2001; Mintzberg et al. 1998).

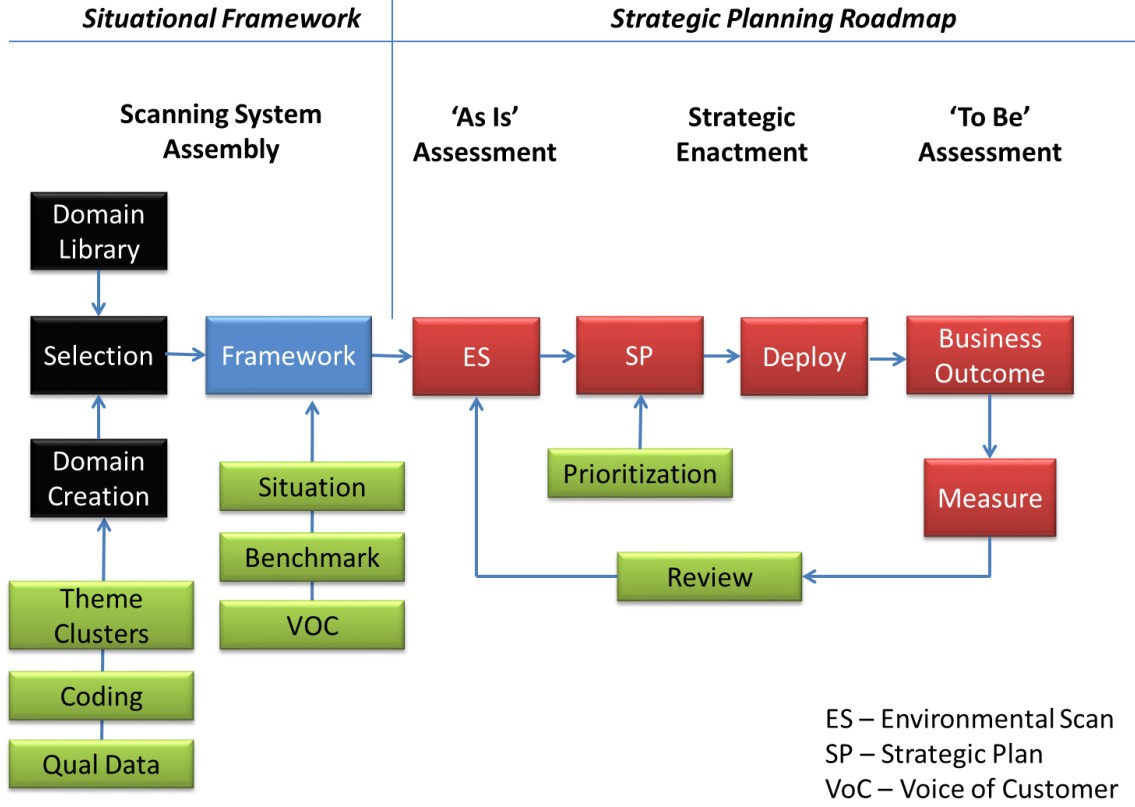

**Figure 6.** Strategic planning roadmap.

### 8.1. Task Selection

Task selection within a domain directly impacts the domain specific goal achievement (Bourgeois 1980). These tasks are born out of force field analyses and are aligned with the goals imposed on a situation. Ambiguity, uncertainty, and an understanding of residual risk in a system are critical aspects of environments in transition (Daft and Weick 1984; Wilkinson 2006). Specifically,

perceived environmental uncertainty (PEU) is the difference between the information needed to make a decision about a task and the information available (Galbraith 1973). PEU tends to mask composite measures sought after during scanning activities (Boyd and Fulk 1996) that drive task creation. Concurrently, leaders tend to act on a perceived environment (Boyd et al. 1993) with the goal of achieving a desired adaptation (Davis and Meyer 1998; Hambrick 1981). Task leaders must also know that environmental variation relates to changes that may occur independent of a leader's ability to notice, comprehend, or interpret environment related data (Doty et al. 2006). Regardless, organizations tuned into their environment are more likely to succeed because they are able to respond 'predictively' through meaningful action and contingencies to a wide range of signals (Slaughter 1999).

*8.2. Practitioner's Needs*

Discussions about scanning are relevant to change practitioners involved in strategic execution (Freifeld 2013; Love 2011). At the same time, stakeholders want to minimize frustration and collateral damage during transitions, which would otherwise be debilitating and draining of purpose and passion (Love and Cugnon 2009). To illustrate poor strategic planning performance, research shows that only 17 percent of cross-border mergers and acquisitions (M and A's) create shareholder value. This is attributed to a lack of attention given to non-financial aspects of strategic plans (Calipha et al. 2010; Galpin and Herndon 2007; Lau et al. 2012; Shimizu et al. 2004; Stahl and Voigt 2008; Weber et al. 1996), and an inability of ES efforts to produce useful information (Slaughter 1999). Clearly, it is not only important to ask the right questions (Klein 2004), but it is also important to ask the right people (Chrusciel 2011). While there is literature on situational leadership (Blanchard et al. 2013), there appears to be little information available on creating situational frameworks for the environmental scanning used in strategic planning. Furthermore, there does not appear to be any information on linking scanning data to action plans. The execution of change initiatives affects jobs, the work experience of employees, and their quality of life. Organizations that are not scanning and adjusting regularly succumb to a world that is experiencing dysfunction, stress, and upheaval on an unprecedented scale (Slaughter 1999).

## 9. Implications

Clarity around strategic planning is needed for better organizational outcomes. Through collective sharing, predictive learning, and reflection, change agents can enhance their ES techniques. Quick wisdom generation is needed in a fast-paced environment; however, sometimes these efforts to collect information are hampered by constraints imposed by internal and external sources. For example, the availability of critical information may be a challenge for a planner or decision maker, because of a lack of a business intelligence capability. By increasing the 'speed to wisdom', the strategic enactment potential is heightened. Practically, a business system can be helpful in turning wisdom into action, as long as the data is collected and recorded accurately, and can be extracted in a meaningful format (Choudhury and Sampler 1997; Davis and Olson 1985).

Factors important to achieving an objective can come in a number of forms, such as income or revenue, cost avoidance, risk mitigation, cost reduction, and contract renewal. Threats are factors that might compromise these benefits. Threats also help to promote a sense of urgency to execute a plan. If an effective plan is not exploited, the benefits to the organization are in jeopardy. Key performance metrics that measure and relate to appropriate business outcomes should be listed, defined in description, and defined logically. Ultimately, domain specific metrics contribute to critical success factor reporting, can be aggregated across a framework of domains, and are well displayed in a balanced scorecard (Kaplan and Norton 1996).

## 10. Conclusions

Strategic agility enables an organization to achieve desired outcomes (Morris 2014; Sull 2010) and sustain their competitive advantage (D'Aveni et al. 2010). Potentially strategic agility can be expressed in an algorithm, as follows:

$$\text{Strategic Agility} = \text{Scanning Accuracy} \times \text{Agility} \times \text{Adaptability}$$

To elaborate, scanning accuracy is simply the capability to obtain and exploit the knowledge of an organization's situation in its environment, current and future. Agility is the ability to minimize the negative influence of obstacles on momentum. Adaptability is an organization's ability to transform itself, thereby increasing its situational efficacy (Davis and Meyer 1998).

In some cases, strategic planning is ad hoc with a dependency on serendipity, whch may or may not be forthcoming (Aguilar 1967; Hambrick 1979; Hambrick 1981; Kefalas and Schoderbek 1973). Some organizations, however, see value in planning and execution (Charan et al. 2012). It is also likely that even a mature organization may not appropriately understand or leverage the links between the domains that can improve the outcome potential. Furthermore, a confident organization, inviting of criticism, may allow their customers, internal and external, to influence their framework design, its weighting, and the metrics that are being applied (De Pree 2004). When direct and indirect stakeholders know that a strategic plan is thorough, and when they are given opportunities to influence the plan (Chrusciel 2011), they are more inclined to be cooperative and in alignment. Engaged stakeholders are also more likely to follow a meaningful path laid out for the organization, even if sacrifice is involved (De Pree 2004); even so, it is better to achieve a goal through strategy than through sacrifice. Although complexity is intensified with diverse stakeholders, the methods discussed in this article aid in the efficient and timely accomplishment of the organizational transitions necessary in turbulent and evolving markets (Aguilar 1967; Buchholtz and Kidder 1999; Choo 1999; El Sawy 1985; Kefalas and Schoderbek 1973; McEwen 2008).

The author has taken the ontological position, based on literature reviews, that the bridge does not exist between ES and strategic enactment, or that it has minimally not been developed. The practical and theoretical implications of this article, then, are that it has illustrated that a link can be created between an appropriately designed environmental scanning tool and an action plan. This bridge is typically not present and so value is not added by conducting a scan. The method outlined in the article enables the reader to not only execute the bridging, but also allows for a better understanding of the forces at play in the scenario. Prioritization is enabled by understanding the impact of the elements in the plan. Furthermore, the reader is able diminish the forces that are holding back the changes that are needed. The approach taken will be a significant driver for success. By removing the roadblocks that these forces against the change are applying, momentum can be created and change fluidity occurs. This article also illustrated and explained the social dynamics at play, along with the interconnectedness between environmental domains. The author hopes that this article has added to the epistemological understanding of the value added by environmental scanning, by providing additional knowledge elements and by increasing clarity and capability.

## 11. Limitations

This article has argued that robustness of strategic planning can be enhanced by deploying a situational framework for environmental scanning that exploits dynamic sense-making mechanisms. A lack of understanding around domain libraries, domain creation from coding, framework assembly, relationships between domains, and planning strategically within domains, including measurements, points to further research being needed. Additional research also can be done on domains and their connectedness, helping practitioners create and better understand domain complexity. More needs to be understood and written about attributes and appropriate segmentation of domains into sub-domains, including their associated metrics. Clearly, more needs to be written about the nature of



and the mapping of the internal and external domain forces, and their influence on the networks in which they reside. Additional study could be done to understand the constraints on domain specific planning activities. Finally, controls around a framework assembly process to maintain focus, enable stakeholder mobilization, and avoid distractions, needs to be understood. This is compelling because an ability to customize an environmental scanning framework promotes strategic enactment, and the realization of desired tangible and intangible outcomes.

**Funding:** This research received no external funding.

**Conflicts of Interest:** The author declares no conflict of interest.

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
