# Peer review of "Assembling Frameworks for Strategic Innovation Enactment: Enhancing Transformational Agility through Situational Scanning"

_admsci, doi:10.3390/admsci8030037_

Round 1
Reviewer 1 Report
BROAD COMMENT
This paper is very interesting but he needs an important update, in particular about the literature review. The large part of the articles used in the literature review are too old. The author could use a newest literature about this very interesting topic. Also, the author need to change the titles of the subparagraphs because they are not clear. A better explication of the research questions maybe could improve the quality of the work. The conclusions must be improved. In the end, regard the general level of the paper, with these integrations he could be an high quality research.
SPECIFIC COMMENT
LINE 31
You need to explain in which way the innovation impact on the network. For example, you could talke about financial and economic performance in international market.
You could use these papers:
Venturelli, A., Caputo, F., & Pizzi, S. (2018). L’impatto del contratto di rete nei processi di internazionalizzazione: alcune evidenze empiriche sulle PMI italiane. MANAGEMENT CONTROL.
Li, J., Xia, J., & Zajac, E. J. (2018). On the duality of political and economic stakeholder influence on firm innovation performance: T heory and evidence from C hinese firms. Strategic Management Journal, 39(1), 193-216.
Zaefarian, G., Forkmann, S., Mitręga, M., & Henneberg, S. C. (2017). A capability perspective on relationship ending and its impact on product innovation success and firm performance. Long Range Planning, 50(2), 184-199.
LINE 109
Why you don't talk about "strategic alliances" or "coopetition"? Read some paper about these topic. I think could be useful for your paper.
Author Response
Thank you for your feedback. You have helped me to improve this article.
I have accomplished the following to enhance the article:
Added references in the introduction.
Explicated research questions.
Clarified sub titles
Enhance the conclusion.
Set up limitations section.
Clarified the research gap.
Included objective of the paper.
The organization of the paper is in the second part of the explication of research questions.
Clustered research questions to narrow paper scope. Removed two questions.
Improved practical and theoretical implications. Added paragraph.
Explained methodology of paper.
Checked references and made changes.
Reviewer 2 Report
The paper has some potential but it needs some improvements:
- in the intro I would like to see a better representation of the research gap and of the objective of the paper. Moreover, you can also include the organization of the subsequent parts of the manuscript (organization of the paper).
- try to cluster the research questions, if not the focus of the paper is too wide
- practical and theoretical implications may be improved
- I suggest you to explain the methodology as soon as possible in to the paper
- Check the references (some of them are not included or incorrect)
Author Response

(The authors gave the same response as above.)

Reviewer 3 Report
This paper is very interesting and novel and I think it could be published in Administrative Sciences, but the authors might consider the following question before publication:
-The objectives of the paper would be more clarified if the work had propositions. I believe that the hypotheses could be considered with the variables that the authors use
-A table that includes similar studies carried out to date could improve the framework
-The future research are not included in the text
Author Response

(The authors gave the same response as above.)

Round 2
Reviewer 2 Report
Good improvements. One last comment. From the title and keywords "agility" should be an important word within the paper. However, only 2 times has been cited. Make it consistent and clearer.
Author Response
I have increased the references in the document regarding 'agility' from 11 to 18 by adding 3 new sources.